# Chemopreventive effect of pomegranate and cocoa extracts on ultraviolet radiation-induced photocarcinogenesis in SKH-1 mice

**Francisco José Gómez-García[1], Antonia López López[1], Yolanda Guerrero-Sánchez[1] \*, Mariano Sánchez Siles[1], Francisco Martínez Díaz[2], Fabio Camacho Alonso[1]**

**1** Department of Dermatology, Stomatology, Radiology and Physic Medicine, Faculty of Medicine, Campus of Excellence Mare Nostrum, University of Murcia, Murcia, Spain, **2** Department of Pathology, Hospital General Universitario Reina Sofía, Murcia, Spain

\* yolanda.guerreros@um.es

**Data Availability Statement:** All data are included in the paper.

## Abstract

Non-melanoma skin cancer (NMSC) has a high and increasing incidence all over the world. Solar radiation is the main aetiology for humans. Although most research into photocarcinogenesis uses UVB as a source of radiation, UVA is also carcinogenic in long term. Pomegranate (PGE) and cocoa (CE) extracts have been used for medicinal purposes for time immemorial. Recently, it has been claimed that some of their properties may be an effective preventative measure against photocarcinogenesis and photoaging, but to date in vivo models have not been tested using RUVA, the objective of the present work. A lower incidence of lesions was observed in SKH-1 mice treated with PGE (p<0.001), and lower incidence of invasive squamous carcinoma in both treatment groups (p<0.001 for PGE and p<0.05 for CE); the PGE group also showed a lower level of cell proliferation than the control group (p<0.001). Significantly greater p53 alteration was observed in the control group than the treatment groups (p<0.001 for PGE and p = 0.05 for CE). No significant differences were found in relation to TIMP-1 and MMP-9. Taken together, the results suggest that oral feeding of PGE and CE to SKH-1 mice affords substantial protection against the adverse effects of RUVA, especially PGE.

## Introduction

The incidence of skin cancers exceeds all other malignant neoplasias together. Among these, the most aggressive is melanoma. Although it only represents 5% of all skin cancers, melanoma causes 80% of all deaths from skin cancer (Weinstock 2008) [1]. Non-melanoma skin cancer (NMSC) has a high incidence all over the world, with the number of cases increasing annually, see [2]. Although most are basal cell carcinomas (80%), which have an excellent prognosis, between 15 and 25% are squamous cell carcinomas (SCC), with worse prognosis, whereby they are associated with not inconsiderable rates of morbidity and mortality (Eisemann et al. 2014) [3]. In the U.S.A. incidence varies between 5 and 499 cases per 100.000 inhabitants depending on latitude. Its annual increase has been estimated at between 50 and 200% over

**Funding:** Yolanda Guerrero-Sánchez is partially supported by Ministerio de Ciencia, Innovación y Universidades grant number PGC2018-097198-B-I00 and Fundación Séneca de la Región de Murcia grant number 20783/PI/18.

**Competing interests:** The authors have declared that no competing interests exist.

the last three decades, and these figures continue to increase due to aging populations in the developed world (Waldman and Schmults, 2019) [4]. Although the rate of hospitalization due to melanoma is higher, the incidence of NMSC makes health care costs higher for this type of skin cancer in absolute terms (Duarte et al. 2018) [5]. Exposure to sunlight, pale skin, and immunosupression are risk factors for NMSC. Solar radiation is the main cause of this type of cancer in humans, particularly those with white skin. Ultraviolet A (UVA) radiation (320-400 nm) is the main component of solar radiation arriving on the Earth's surface (90-99%) and is associated with both benign and malignant skin cancers. While UVB provokes more direct damage to cellular DNA, the effects of UVA radiation are indirect, derived from oxidative damage caused by singlet oxygen release, which provokes diverse genetic alterations which vary from point mutations to crude chromosomal dislocations (Bachelor and Bowden, 2004) [6].

*Punica granatum L.* (pomegranate) is a deciduous shrub which has been extensively used in traditional medicine in many regions of the world, including the Mediterranean area, Middle East, and central Asia (Shaygannia et al., 2015) [7]. Pomegranate fruit has biological effects including anti-inflammatory and antibacterial activities. It also contains flavonoids (kaempferol, luteolin and naringenin) and alkaloids (caffeic, gallic and ferulic acids, punicalagin, quercetin or catechin), which can prevent carcinogenesis both in vitro (Singh et al., 2002) [8] and in vivo (Justin et al., 2003) [9]. The seeds of the plant *Theobroma cacao L.* (Sterculiaceae) are the raw material from which one of the most widely consumed functional foods in the world is made–cocoa. The Mayas and the Aztecs used cocoa for both nutritional and curative purposes (Scapagnini et al., 2014) [10]. Cocoa extract is rich in polyphenols, mainly catechins and proanthocyanidin flavanols, but also gallocatechin and epicagallocatechin or methylxanthine compounds (theobromine and caffeine) among other components with reactive oxygen species (ROS) scavenging properties (Bosch et al., 2015) [11]. The quantity and proportion of these components is highly variable due to differences in processing and manufacturing. Although an increasing number of published papers have presented evidence that cocoa extract contributes to endogenous photoprotection and has anti-aging and anti-wrinkle effects, little research has investigated its role in photocarcinogenesis (Kim et al., 2016) [12].

SKH1/CRL mice are an excellent animal model for use in photodamage studies. Their skin is very sensitive to ultraviolet radiation; they are euthymic, immunocompetent, and easy to look after. Oral feeding of both pomegranate fruit and cocoa extracts to SKH-1/CRL mice has demonstrated substantial protection from the adverse effects of UVB radiation derived from modulation of early biomarkers of photocarcinogenesis (Afaq et al., 2010) [13], but this has not been proved using UVA radiation in vivo. In this context, the main objective of the present study was to evaluate the protective effects of pomegranate and cocoa extracts during UVA-induced skin cancer in SKH-1 hairless mice. See for other approaches [14, 15] or [16].

## Materials and methods

### Chemicals

The extracts were supplied by Naturex S.A. (Avignon, France). Pomegranate extract (reference EA140914, batch no D047/002/A11) was obtained from Punica granatum L. rinds (plant extract ratio between 5:1 and 7:1). The extraction solvent contains 70% ethanol, 30% water. The nutritional composition of the extract was as follows: energy 367 kcal/100 g; carbohydrates 84% (w/v); fat 1.4% (w/v); fiber 3.8% (w/v); proteins 2.4% (w/v), and total specific phenolic 12.62% (w/v)with, respectively, 2.77%; 7.92%, and 1.93% (w/v) ofpunicalagin A, punicaligin B, and ellagic acid. Cocoa extract (batch no TCP 19013) was obtained from Theobroma cacao L. complete seeds with the method patented by the company (EP2071961A1). The resulting

product is a powdered semi-purified extract with a total polyphenol content of 20–45% and an organic fraction, from which, after concentration and drying thereof, a powdered purified extract is obtained with a total polyphenol content of 60–90%. The content of flavonols flavanol monomers catechin (24.2) and epicatechin (115) and 226.6 of total oligomeric procyanidins, flavonols (quercetin-3-O-glucoside 1.0; quercetin-3-O-arabinoside 1.0), anthocyanins and alkaloids (10.3 theobromine and 1.1 caffeine).

The extracts were supplied by Naturex S.A. (Avignon, France).

## Animals and treatment

The study used 27 SKH-1/CRL female mice, four weeks old at the beginning of the experiment. The animals (euthymic and immunocompetent) were obtained from Charles River Laboratories (Wilmington, MA, U.S.A.) and acclimatized for 10 days at the Animal Laboratory Service of the University of Murcia (Spain) before the start of the experiment. They were randomly distributed into three groups. Group I (n = 9) were exposed to UVA radiation. Group II (n = 9) animals were also treated with PG, which was orally administered in the diet. Group III (n = 9) was treated with UVR and cocoa extract with ad libitum access. Complete experimental groups (9 animals) were housed in methacrylate cages transparent of $40 \times 30$ cm, covered with metal right and with space for food and water.

The standard chow (PanLab, Barcelona) had 14.3% protein, 4.0% fat, 48.0% carbohydrate, 4.1% crude fiber, 18.0% neutral detergent fiber and 4.7% ash; energy density, 2.9 kcal/g). The doses corresponded to 520 and 228 mg of extract per kg body weight per day of pomegranate and cocoa extract respectively, approximately 41.9 and 18.37 mg per kg body weight HED. The doses of the agents were selected in order to provide equivalent concentrations with other anticancer researches where has been previously tested the positive effect of these compounds (Vlachojannis et al., 2015) [17]) and Urpi-Sarda et al., 2009) [18].Dietary intervention lasted for 29 weeks (including 2 weeks of pretreatment). Body weight, food and water intake were recorded throughout the study. The animals were housed in climate controlled rooms (24˚C at 50% humidity) with 12-hour light/dark cycles and free access to food and water, and treated in accordance to the rules of the European Union for the protection of animals used for experimentation (2010/63/EU). All experimental protocols comply with the ARRIVE guidelines and were approved by the Bioethics Committee of the University of Murcia.

## UV irradiation

The mice were subjected to UV irradiation using a Philips lamp (Type HB 554/01/A) with 8 Philips Performance S 100W tubes. The lamp has an emission spectrum of 220-425 nm with a maximum peak of 364 nm (98.6% UVA and 1.4% UVB). The animals were exposed to radiation three times a week for a total of 80 sessions, each lasting 60 minutes (always at the same hour in the morning). For this purpose, the animals were placed in PVC cages, with individual separators and a metallic lattice ceiling, which were placed under the ultraviolet lamp with a distance of 20 cm between the light source and the skin. The energy received in each session was 21.1 J/cm2, so that at the end of the experiment the total energy received by each mouse was 1688 J/cm2.

## Macroscopic study

A detailed macroscopic study of the dorsal skin was made after each session and digital photographs were taken on millimeter paper to measure the lesional areas at the end of the experiment (80 sessions). An application of the image processing and analysis software platform

Leica Qwin was used to obtain a binary mask of the lesional areas, which were measured individually.

## Microscopic study

At the end of the experiment, animals were sacrificed by $CO_2$ overdose. A necropsy was performed on the dorsal skin and viscera. The samples collected were embedded in paraffin and stained with haematoxylin and eosin for microscopic analysis. The presence of normal skin, actinic keratoses, dysplasia, carcinoma in situ and invasive carcinoma was examined under light microscopy (Leica DM 6000B). Immunohistochemical study of tumors was conducted by the Pathology Department at the Reina Sofía University Hospital (Murcia, Spain) using the following antibodies: proliferating cellular nuclear antigen (PCNA); p53; anti-metaloproteinase-9 (MMP-9); and anti-metallopeptidase inhibitor 1 (TIMP-1) (Dako, S.A, Barcelona, Spain). Immunohistochemical staining was performed using the labeled streptavidin-biotin method. The sections were incubated overnight at 4˚C with antibodies to PCNA (PC-10; Dako Corporation, Glostrup, Denmark); MMP; and TIMP-1 at a 1:200 dilution. Negative controls were treated with all reagents except the primary antibody. PCNA expression was quantified by means of image analysis. Firstly, the prepared specimens were digitalized by high resolution microscopy scanning with the Leica SCN400F. Then using an image analysis module from the Slidepath Digital Image Hub, three 100 μm2 areas were traced on each image. The areas selected were the tumor invasion front (in the case of tumors) and the epidermis basal layer when no carcinoma was present. The software produced a count of the number of positive nuclei and the total number of cells, using the Nuclear Algorythm 3.0, which detects PCNA marking by means of a previously defined color pattern. Inmunohistochemical quantification (PCNA and P53) was expressed as the mean percentage of PCNA and P53 immunohistochemical positive cells, in relation to the total number of cells (Ibrahim and Elwan, 2017) [19]. In semi-quantitative immunohistochemical analysis (TIMP, and MMP-9), scores were calculated by multiplying the graded intensity of stained stroma (grade 0-3) by the percentage of stained stroma surface (grade 0: no stained stroma, grade 1: less than 10% stained stroma, grade 2: 10-50% stained stroma, and grade 3: 51-100% stained stroma) (Preidl et al., 2019) [20]. All microscopic analyses were carried out by two pathologists blinded to the results of the study.

## Statistical analysis

For the statistical analysis, we have used the SPSS software, version 20.0 (SPSS [R]Inc., Chicago, IL, USA). A descriptive study of each involved variable was performed. The associations between qualitative variables were determined by Pearson's Chi-square test. For the quantitative variables, the ANOVA and Tuckey tests were performed, determining in each case if the variances were homogeneous. Differences were regarded as significant if $p \leq 0.05$ and highly significant if $p \leq 0.01$.

## Results and discussion

Skin cancer is more common among the white races and has undergone a dramatic increase during the last 40 years, due to social changes with regard to exposure to sunlight and UV radiation (UVR) (Weinstock, 2008) [1]. As a result, UVR has come to be regarded as the most important environmental carcinogen, and has driven the health-care sector on a quest for better information and management regarding the effects of solar radiation (Duarte et al., 2018) [5]. The UV spectrum can be divided into three wavelength ranges, UVA (320–400 nm), UVB (280–320) and UVC (200–280 nm). Of the solar radiation that reaches the surface of the earth, more than 90% is comprised of UVA and 1-10% is comprised of UVB. A large portion of UVB

and all of the UVC wavelengths are absorbed by the ozone layer. So the biological effects of solar radiation are attributed to UV A and B radiations. For many years, it has been known that UVB is a complete carcinogen that can generate squamous-type carcinomas in animals. Its main biological target is DNA, where it produces CC to TT or C to T point mutations (Bosch et al., 2015) [11]. So, as far as most authors have been concerned, the spectrum in which skin carcinomas are produced has been the UVB region, attributing less carcinogenic activity to the UVA region (Forbes, 1981) [21]. Forbes considered that while UVB played a key role in initiating tumors, UVA energy intervened in promoting it. The current understanding is that although UVB is approximately 1,000-10,000 times more carcinogenic per $J/m^2$ than UVA, chronic UVA radiation alone is also carcinogenic as shown in diverse in vivo experiments since the nineteen nineties (De Laat et al., 1997) [22]; (de Gruijl, 2004) [23]. Nevertheless, as tumor genesis is faster with UVB, most research into UV-related pathogenesis and particularly those works investigating preventative or palliative treatment have used UVB radiation sources (Kim et al., 2016) [12]; (Rigby et al., 2017) [24]; (Afaq et al., 2010) [13]; (Park et al., 2010) [25]. The equipment employed in the present study was a cosmetic tanning lamp that emitted 98.6% UVA and 1.45% UVB radiation. UVA radiation acts directly by generating ROS, which act on light-sensitive molecules such as porphyrins and nicotinamide adenine dinucleotide hydrate (NADH). One important aspect is that, while UVB radiation is almost

| Study groups cancer | Normal skin | Actinic Keratoses | Dysplasia | Carcinoma in situ | Invasive |
| --- | --- | --- | --- | --- | --- |
| | n (%) | n (%) | n (%) | n (%) | n (%) |
| Control | 0 (0)§ | 0 (0)† | 0 (0) | 0 (0) | 9 (100)§,† |
| *Punica granatum* | 7 (77.78)#,† | 1 (11.11) | 0 (0) | 0 (0) | 1 (11.11)# |
| | p<0.001 | | | | p<0.001 |
| Cocoa cathechin | 0 (0)§ | 4 (44.45)# | 1 (11.11) | 2 (22.22) | 2 (22.22)# |
| | p<0.05 | | | | p=0.001 |

# Significant difference compared with Control group; § Significant difference compared with *Punica granatum* group; † Significant difference compared with Cocoa cathechin group

Fig 1. Table of comparison of incidence of skin lesions between study groups (Pearson's $\chi^2$ test).

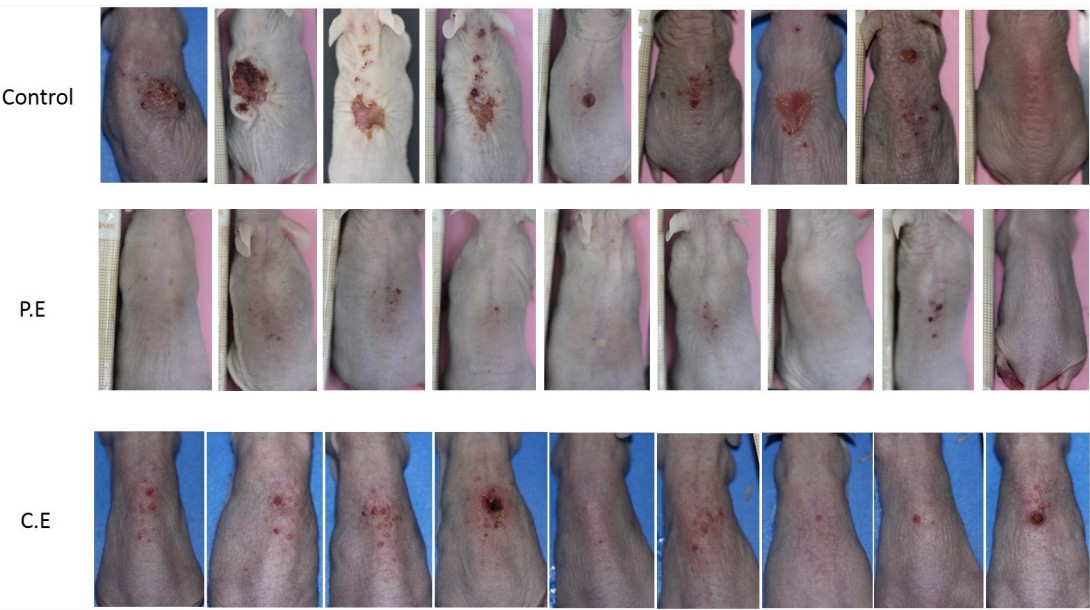

**Fig 2. Skins of animals at the end of the irradiation period (80 sessions).** Control, PE (pomegranate extract) and CE (cocoa extract).

completely absorbed by the epidermis (Seebode et al., 2016 [26]), UVA radiation is able to reach the dermis layers and even affects circulating blood cells (Möller et al., 2002) [27]. Among other effects, UVA radiation can inhibit DNA repair and induce metalloproteinase synthesis of the cell matrix (MMP), which can boost the skin cancer's biological aggression (Fisher et al., 2001) [28]. The present study saw a 100% incidence of spinocellular carcinomas in the Control Group (Fig 1), which highlights the fact that UVA radiation is capable of provoking the entire spectrum of photoaging lesions. After their first exposure to UVA, all animals

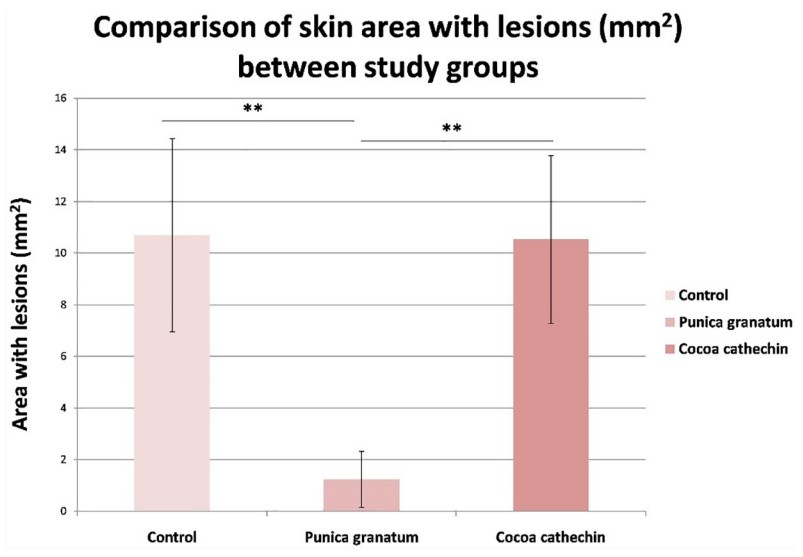

**Fig 3. Comparison of macroscopic morphometric analysis of skin area with lesions (mm$^2$) between study groups (Tukey test).** Significant difference ($^*p \leq 0.050$; $^{**}p < 0.010$).

in the Control Group presented temporary diffuse erythema on their back, which disappeared or diminished 2 or 3 hours after exposure to radiation. After the tenth to twelfth session, the erythema became permanent accompanied by a markedly geometric skin pattern characterized by a reticular appearance, which became more evident over time. The appearance of prominent longitudinal wrinkles along the whole surface of the back was remarkable, presenting skin laxity and loss of elasticity, which progressively increased in length and thickness alternating with more local areas that were slightly raised and of irregular aspect, generally accompanied by a squamous surface. These areas were irregularly located between the wrinkles and from that time until the end of the UVA sessions the skin presented increasingly extensive areas of telangiectatic appearance. After the fortieth session, the skin took on a diffuse granular appearance, and around the fiftieth session, suffered extensive irregular surface ulceration. In control specimens, neoplastic lesions started to appear after the fiftieth session, while in the treatment groups, macroscopic lesions were not detected until after the sixtieth session. These lesions were of nodular type and attached to deep tissue planes. Some of them had irregular edges, were raised, with congestive borders and a verrucous appearance, while others were depressed in the center, of squamous appearance, with surface ulceration. These were dirty-based ulcers, which in some cases presented spontaneous bleeding. By the end of the 75 sessions of the study, all the control group animals developed lesions of neoplastic appearance, with similar lesions appearing in the group treated with cocoa extract (Fig 2). Numerous studies have backed the hypothesis that the use of plant-derived antioxidant substances has a favorable effect in the treatment or prevention of photoaging and photocarcinogenesis. Diverse studies have demonstrated reductions in inflammatory response, oxidative stress, DNA damage, the early appearance of erythema, premature wrinkling, or skin cancer after prolonged exposure to UVR as a result of treatment with natural compounds (Bosch et al., 2015) [11]. In the present study, we used purified extracts of cocoa and pomegranate, observing significant reductions in the incidence of skin carcinoma in both groups, but especially in the PGE group. In the latter group, hardly any of the animals presented lesions, their

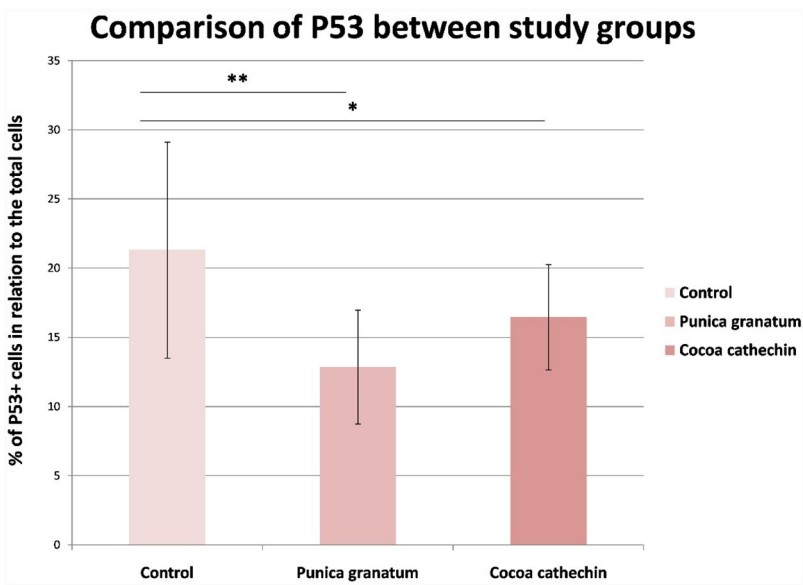

**Fig 4. Comparison of percentage of P53 immunohistochemical positive cells in relation to the total number of cells between study groups (Tukey test).** Significant difference ($^*p \leq 0.050$; $^{**}p < 0.010$).

skins being classified as normal in histopathological analysis (Fig 1). No previous in vivo trials of cocoa extract have been conducted, although (Kim et al. 2016) [12] published an article in which it was shown that oral cocoa supplements protected against the formation of wrinkles provoked by UVB radiation by regulating the genes that intervene in the formation and degradation of the extracellular collagen matrix (cathepsin G and serpin B6c) (Kim et al., 2016) [12]. The pomegranate has been studied more often, although mainly in relation to UVB radiation. Pomegranate extract is a high source of ellagitannins, anthocyanins, and tannins and possesses a powerful antioxidant action (Bassiri-Jahromi, 2018) [29]; (Afaq et al., 2010) [13] carried out a trial using SKH-1 hairless mice treated with PGE in drinking water (0.2%, wt/vol) for 14 days before irradiation and then exposed to UVB radiation only once (180 mJ cm2). The results showed inhibition of: skin edema, hyperplasia, infiltration of leukocytes, lipid peroxidation,

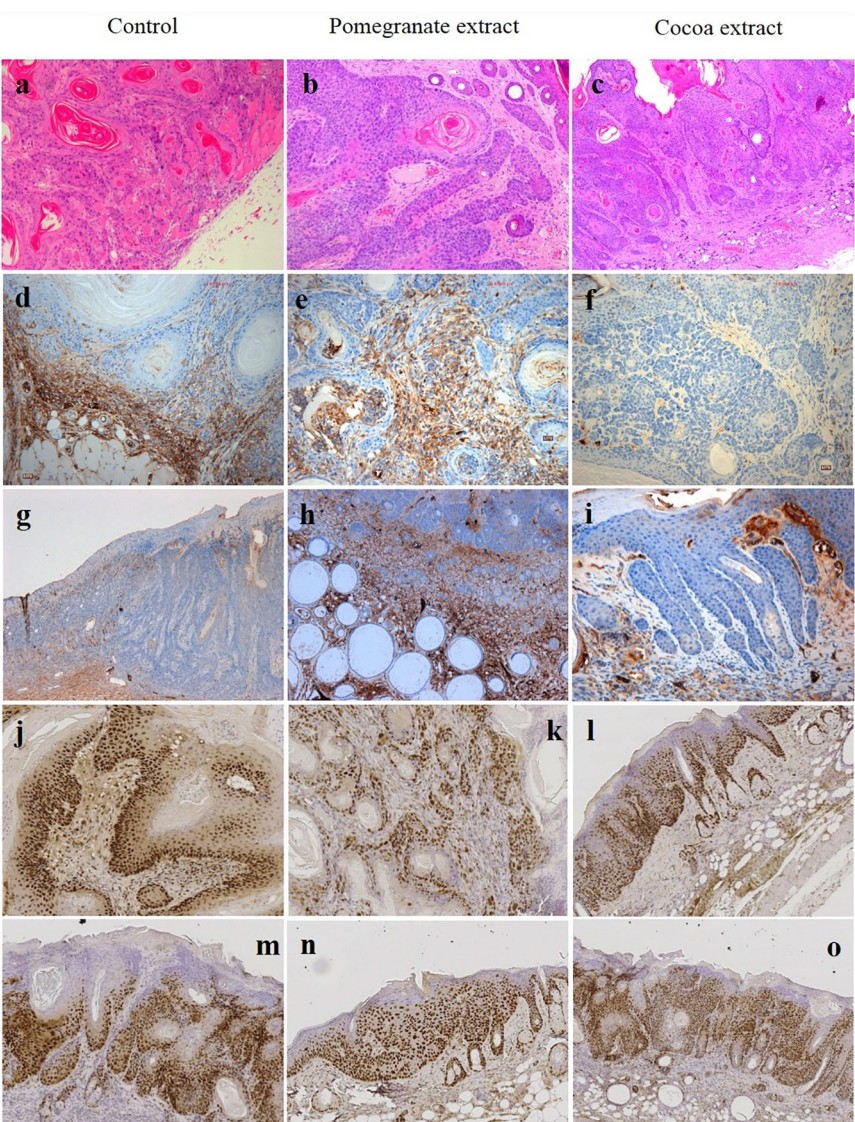

**Fig 5. Spinocellular carcinomas in the different study groups (Haematoxilin-eosin) (a,b,c); MMP and TIMP-1 expression in surrounding tumour stroma (d, e, f and g, h, i respectively); p53 (j, k, l) and PCNA (m, n, o) immunostaining.**

hydrogen peroxide generation, ornithine decarboxylase and cyclooxygenase-2 activity (Afaq et al., 2010) [13]. In particular, following UVA radiation, PGE inhibits UVA-mediated phosphorylation of STAT3, AKT, ERK1/2, mTOR and p70S6K, decreases up-regulation of PCNA and Ki-67 expression, and up-regulates Bax and Bad expression in human keratinocytes (Syed et al., 2005) [30], inhibiting the effects of photoaging (Silva et al., 2019) [31]. But its effects against UVA radiation have not been investigated in vivo. In the present study, we observed a clear decrease in macroscopic lesions, particularly in the PGE group, with statistically significant differences in comparison with the control and cocoa groups, which did not show significant differences between them (Fig 3). In microscopy analysis with hematoxylin and eosin staining, the incidence of squamous cell carcinoma was 100% in the control group, while animals treated with PGE only developed one carcinoma (11.11%) and those treated with CE developed four (22.22%). Statistically significant differences were found between both the treatment groups and the control. The very slight skin affection in seven of the nine animals treated with PGE was remarkable, their skin remaining normal (Fig 1).

The protein p53 has diverse and complex functions, including cell cycle regulation. Its alteration is directly related to cancer development, as it mutates in 50% of human cancers, including NMSC (Benjamin and Ananthaswamy, 2007) [32]. In SKH-1 mice chronically exposed to UVB radiation, inactivation of p53 mutations has been reported to occur in 50-70% of tumors, this being an early event, detected as early as 1 week post-UVB exposure (Rigby et al. 2017) [24]. The present study found lower mutated p53 expression in the treatment groups than in the control, with statistically significant differences, but without any significant difference between PGE and CE (Figs 4 and 5). After UVB exposure, oral treatment with PGE, has also shown a positive effect on the expression of tumor suppressor p53 and cyclin kinase inhibitor p21(Afaq et al., 2010) [13].

Proliferating cell nuclear antigen (PCNA) is an auxiliary protein necessary for DNA synthesis and repair. PCNA quantification is used as a marker of cell proliferation in tissues to assess the efficacy of chemopreventative drugs in cancer research (Smolarek et al., 2014) [33], and is

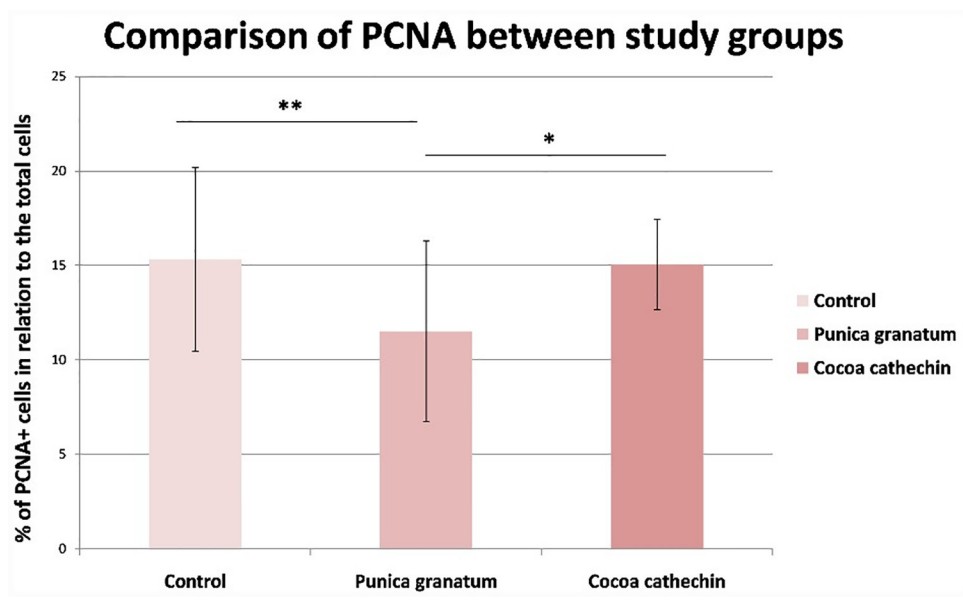

**Fig 6. Comparison of percentage of proliferating cell nuclear antigen (PCNA) immunohistochemical positive cells in relation to the total number of cells between study groups (Tukey test).** Significant difference (*p≤0.050; **p<0.010).

considered an independent factor in the prognosis of survival, as its expression is seen to increase in lesions with high malignancy in both NMSC and melanoma (Ruksha et al., 2007) [34]. In the present study, PCNA expression was similar in the control and CE groups but lower cell proliferation was observed in the PGE group with significant difference in comparison with the other two groups (Figs 5 and 6). These results concur with other studies of human keratinocytes irradiated with UVA (Syed et al., 2005) [30], and of SKH-1 mice irradiated with UVB (Afaq et al., 2010) [13].

Matrix metalloproteinases (MMPs) are a family of proteolytic enzymes capable of degrading different components of the extracellular matrix (ECM). Their role in healthy tissue is their physiological remodeling through both formation and repair. In cancer, they are related to important processes such as angiogenesis, cell proliferation, cellular invasion, and metastasis. MMP-9 belongs to a subgroup of gelatinases, and its function is the degradation of

| Study groups | TIMP | MMP-9 |
| --- | --- | --- |
| | mean ± SD | mean ± SD |
| Control | 0.77 ± 0.43 | 1.00 ± 0.00 |
| *Punica granatum* | 1.44 ± 0.72 | 1.33 ± 0.50 |
| Cocoa cathechin | 1.67 ± 0.71 | 1.44 ± 0.72 |
| * **SD = standard deviation. # Significant difference compared with Control group; § Significant difference compared with *Punica granatum* group; † Significant difference compared with Cocoa cathechin group** | | |

**Fig 7. Table of comparison of degree of intensity (0–3) of stained stroma with TIMP and MMP-9 between study groups (Tukey test).**

basement membrane components such as type IV collagen and laminin, as well as other ECM components (Andisheh-Tadbir et al., 2016) [35]. This process is regulated by tissue inhibitors of MMPs (TIMPS). Although there are four types, one of the most employed in NMSC is TIMP-1, which is usually inhibited in patients with skin cancer (Fu et al., 2012) [36]. In the present study, MMP-9 expression was higher in the treatment groups than the control group, even though tumors were more frequent in the latter group. But TIMP-1 showed bigger value in the treatment groups, and so it would appear that inhibition of ECM degradation tended to be more active in these groups, although without reaching statistical significance differences (Figs 5 and 7).

Although in many studies, MMP-9 expression is higher in precancerous tumors and lesions than healthy tissue (Gupta et al., 2014) [37] (Fu et al., 2012) [36] (Andisheh-Tadbir et al., 2016) [35], others such as (Poswar et al., 2013) [38] have found greater expression in microinvasive carcinomas than in those that invade in depth, concluding that MMP-9's proteolytic activity takes place during the initial stages of carcinogenesis and decreases afterwards (Poswar et al. 2013) [38]. This could explain the present study's observations, whereby control group tumors were more evolved than in the treatment groups.

While much research has focused upon the effects of UVB radiation, little is known about UVA-induced photocarcinogenesis in vivo. This lack was the motivation for the present study, which also set out to treat animals with extracts whose efficacy in UVB-induced skin cancer chemoprevention appears to have been demonstrated. Taken together, our results suggest that oral feeding of PGE and CE to SKH-1 mice affords substantial protection from the adverse effects of UVA radiation, especially PGE. Obviously further research is needed to investigate in detail the mechanisms of action of these substances, which could play an important role in NMSC prevention in the future.

## Supporting information

**S1 File.**
(PDF)

## Author Contributions

**Investigation:** Francisco José Gómez-García, Antonia López López, Yolanda Guerrero-Sánchez, Mariano Sánchez Siles, Francisco Martínez Díaz, Fabio Camacho Alonso.

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
