## [Decision Letter · Decision Letter 0]

29 Jan 2020

PONE-D-19-31596

Chemopreventive effect of pomegranate and cocoa extracts on ultraviolet radiation-induced photocarcinogenesis in SKH-1 mice

PLOS ONE

Dear Dra. Guerrero-Sánchez,

Thank you for submitting your manuscript to PLOS ONE. After careful consideration, we feel that it has merit but does not fully meet PLOS ONE’s publication criteria as it currently stands. Therefore, we invite you to submit a revised version of the manuscript that addresses the points raised during the review process.

We would appreciate receiving your revised manuscript by Mar 14 2020 11:59PM. To enhance the reproducibility of your results, we recommend that if applicable you deposit your laboratory protocols in protocols.io, where a protocol can be assigned its own identifier (DOI) such that it can be cited independently in the future. For instructions see: http://journals.plos.org/plosone/s/submission-guidelines#loc-laboratory-protocols

We look forward to receiving your revised manuscript.

Kind regards,

J. Alberto Conejero

Academic Editor

PLOS ONE

Journal Requirements:

2. As part of your revision, please complete and submit a copy of the ARRIVE Guidelines checklist, a document that aims to improve experimental reporting and reproducibility of animal studies for purposes of post-publication data analysis and reproducibility: https://www.nc3rs.org.uk/arrive-guidelines. Please include your completed checklist as a Supporting Information file. Note that if your paper is accepted for publication, this checklist will be published as part of your article.

3. We noticed you have some minor occurrence(s) of overlapping text with the following previous publication(s), which needs to be addressed:

https://doi.org/10.1111/jop.12435

In your revision ensure you cite all your sources (including your own works), and quote or rephrase any duplicated text outside the Methods section. Further consideration is dependent on these concerns being addressed.

4. We note that you have not given a rationale for the chosen treatment doses. Please elaborate this aspect in the Methods section. Please also justify why group 3 received 'UVR' radiation, if this is a misspelling please correct it.

5. Please include additional information regarding pomegranate and cocoa extracts, such as: product name and number, lot/batch number, full ingredient list, method of chemical characterization, purity and yield. Please also include a description of the look and feel of the substance, and the species name of the cocoa plant used.

7. Please ensure that you refer to Figures 6 and 7 in your text as, if accepted, production will need this reference to link the reader to the figure.

8. Please include a separate title for each table in your manuscript.

Additional Editor Comments:

We have received 3 reports of your work. In one of them, the reviewer points out some concerns respect to the model. Please, try to clarify these points when you submit the revision of your work.

Reviewers' comments:

Reviewer's Responses to Questions

**Comments to the Author**

1. Is the manuscript technically sound, and do the data support the conclusions?

Reviewer #1: No

Reviewer #2: Yes

Reviewer #3: Yes

2. Has the statistical analysis been performed appropriately and rigorously? 

Reviewer #1: I Don't Know

Reviewer #2: Yes

Reviewer #3: Yes

3. Have the authors made all data underlying the findings in their manuscript fully available?

Reviewer #1: No

Reviewer #2: Yes

Reviewer #3: Yes

4. Is the manuscript presented in an intelligible fashion and written in standard English?

Reviewer #1: No

Reviewer #2: Yes

Reviewer #3: Yes

5. Review Comments to the Author

Reviewer #1: In this study, the authors have examined the effect of pomegranate and cocoa extracts on UVA-induced skin cancer in SKH-1 hairless mice. Overall, whereas the studies are conducted properly, the model itself has serious concerns. Specifically, UVA radiation also does not induce skin cancers, and indeed the authors did not find any skin cancer in control group; except some kind of invasion which is not that common for non-melanoma skin cancer. Because of the same issue of using UVA, the effects of the test agents are also not evident, though they have shown efficacy against UVB-induced skin cancer; therefore, the studies lack both the novelty and appropriate model. Lastly, the molecules analyzed at the the study end are also not properly justified, and in most cases there is hardly any effect.

Reviewer #2: General Comments

This is an excellent article that investigates the protective effects of pomegranate and cocoa extracts during UVA-induced skin cancer in SKH-1 hairless mice. In general, it is very well written. The research question is clear. Robust, well-explored and well-described methods were used.

Authors demonstrated that 100% of control animals presented spinocellular carcinomas which highlights the fact that UVA radiation can provoke the entire spectrum of photoaging lesions. Animals that received purified extracts of cocoa and pomegranate showed significant reductions in the incidence of skin carcinoma in both groups. In addition, lower mutated p53 expression in the treatment groups than in the control as well as, control group demonstrated higher PCNA and MMP-9 labelling. The present study is a step toward better understanding the UVA photocarcinogenesis and new chemopreventive strategies. However, some changes in the article are needed.

I have the following suggestion and criticism.

- Please add the reference of GLOBOCAN report. Check the incidence of skin cancer reported on it.

Bray F, Ferlay J, Soerjomataram I, Siegel RL, Torre LA, Jemal A.Global cancer statistics 2018: GLOBOCAN estimates of incidence and mortality worldwide for 36 cancers in 185 countries.CA Cancer J Clin. 2018 Nov;68(6):394-424. doi: 10.3322/caac.21492.

- Separate the introduction into three paragraphs to make it easier to follow.

2nd- Punica granatum…

3rd- SKH1/CRL…

- Materials and methods

UV irradiation- Please inform the reference for the protocol used.

- At the end of the experiment, animals were sacrificed by CO2 overdose. Why did you use CO2 chamber? Nowadays, it is recommended euthanasia using an isoflurane anesthetic overdose .

-Results- Please change this title for Results and Discussion

Please separate the following information in new paragaraph-

“ The protein p53 has diverse and complex 227 functions, including cell cycle regulation. Its alteration is directly related to cancer 228 development, as it mutates in 50% of human cancers, including NMSC (Benjamin and 229 Ananthaswamy, 2007) [5]. In SKH-1 mice chronically exposed to UVB radiation, 230 inactivation of p53 mutations has been reported to occur in 50-70% of tumors, this 231 being an early event, detected as early as 1 week post-UVB exposure (Rigby et al. 232 2017) [25]. The present study found lower mutated p53 expression in the treatment 233 groups than in the control, with statistically significant differences, but without any 234 significant difference between PGE and CE (Figures 3 and 5). After UVB exposure, 235 oral treatment with PGE, has also shown a positive effect on the expression of tumor 236 suppressor p53 and cyclin kinase inhibitor p21(Afaq et al., 2010) [1].”

-Please change the following sentence.

“But TIMP-1 showed more expression in the treatment groups, and so it would appear that inhibition of ECM degradation was more 260 active in these groups, although differences did not reach statistical significance (Table 261 2, Figure 5).” If you did not have statistical differences the groups are equals. Rewrite the sentence!

Reviewer #3: The paper with the title Chemopreventive effect of pomegranate and cocoa extracts on ultraviolet radiation-induced photocarcinogenesis in SKH-1 mice is well write and the study design is conducted correctly.

For these reasons the paper is accepted for the publication on PLOS ONE.

6. PLOS authors have the option to publish the peer review history of their article (what does this mean?). If published, this will include your full peer review and any attached files.

Reviewer #1: No

Reviewer #2: No

Reviewer #3: No

---

## [Author Response · Author response to Decision Letter 0]

18 Mar 2020

Please see the attached file where we have answered in detail to all referee's remarks.

---

## [Decision Letter · Decision Letter 1]

7 Apr 2020

Chemopreventive effect of pomegranate and cocoa extracts on ultraviolet radiation-induced photocarcinogenesis in SKH-1 mice

PONE-D-19-31596R1

Dear Dr. Guerrero-Sánchez,

We are pleased to inform you that your manuscript has been judged scientifically suitable for publication and will be formally accepted for publication once it complies with all outstanding technical requirements.

With kind regards,

J. Alberto Conejero

Academic Editor

PLOS ONE

Additional Editor Comments (optional):

Reviewers' comments:

Reviewer's Responses to Questions

**Comments to the Author**

1. If the authors have adequately addressed your comments raised in a previous round of review and you feel that this manuscript is now acceptable for publication, you may indicate that here to bypass the “Comments to the Author” section, enter your conflict of interest statement in the “Confidential to Editor” section, and submit your "Accept" recommendation.

Reviewer #1: All comments have been addressed

Reviewer #2: All comments have been addressed

Reviewer #3: All comments have been addressed

2. Is the manuscript technically sound, and do the data support the conclusions?

Reviewer #1: Yes

Reviewer #2: Yes

Reviewer #3: Yes

3. Has the statistical analysis been performed appropriately and rigorously? 

Reviewer #1: Yes

Reviewer #2: Yes

Reviewer #3: Yes

4. Have the authors made all data underlying the findings in their manuscript fully available?

Reviewer #1: Yes

Reviewer #2: Yes

Reviewer #3: Yes

5. Is the manuscript presented in an intelligible fashion and written in standard English?

Reviewer #1: Yes

Reviewer #2: Yes

Reviewer #3: Yes

6. Review Comments to the Author

Reviewer #1: The authors have addressed all the concerns related to the use and carcinogenic effects of UVA and UVB, as acceptable responses.

Reviewer #2: The authors performed all the modification suggested.I reccomend the aproval of the mansucript.Best Regards

Reviewer #3: The Authors presented all the revisions requested. For this reason the paper could be accepted for the publication

7. PLOS authors have the option to publish the peer review history of their article (what does this mean?). If published, this will include your full peer review and any attached files.

Reviewer #1: No

Reviewer #2: No

Reviewer #3: No

---

## [Editor Report · Acceptance letter]

15 Apr 2020

PONE-D-19-31596R1 

Chemopreventive effect of pomegranate and cocoa extracts on ultraviolet radiation-induced photocarcinogenesis in SKH-1 mice 

Dear Dr. Guerrero-Sánchez:

I am pleased to inform you that your manuscript has been deemed suitable for publication in PLOS ONE. Congratulations! Your manuscript is now with our production department. 

With kind regards,

on behalf of

Dr J. Alberto Conejero 

Academic Editor

PLOS ONE